# Comprehensive Assessment of Mid-Regional Proadrenomedullin, Procalcitonin, Neuron-Specific Enolase and Protein S100 for Predicting Pediatric Severe Trauma Outcomes

**DOI:** 10.3390/biomedicines11082306

**Published:** 2023-08-19

**Authors:** Rustam Zakirov, Svetlana Petrichuk, Olga Yanyushkina, Elena Semikina, Marina Vershinina, Olga Karaseva

**Affiliations:** 1National Medical Research Center for Children’s Health, 119296 Moscow, Russia; 2Clinical and Research Institute of Emergency Pediatric Surgery and Traumatology, 119180 Moscow, Russia

**Keywords:** children, severe injury, polytrauma, traumatic brain injury, multiple organ failure, outcome prediction, MR-proADM, PCT, NSE, protein S100

## Abstract

The development of multiple organ failure and septic complications increases the cumulative risk of mortality in children with severe injury. Clinically available biochemical markers have shown promise in assessing the severity and predicting the development of complications and outcomes in such cases. This study aimed to determine informative criteria for assessing the severity and outcome prediction of severe injury in children based on levels of mid-regional proadrenomedullin (MR-proADM) procalcitonin (PCT), neuron-specific enolase (NSE), and protein S100. Biomarker levels were measured in 52 children with severe injury (ISS ≥ 16) on the 1st, 3rd, 7th, and 14th days after admission to the ICU. The children were divided into groups based on their favorable (n = 44) or unfavorable (n = 8) outcomes according to the Severe Injury Outcome Scale, as well as their favorable (n = 35) or unfavorable (n = 15) outcomes according to the Glasgow Coma Outcome Scale (GOS). The study also evaluated the significance of biomarker levels in predicting septic complications (with SC (n = 16) and without SC (n = 36)) and diagnosing and stratifying multiple organ failure (with MOF (n = 8) and without MOF (n = 44)). A comprehensive assessment of MR-proADM and PCT provided the highest diagnostic and prognostic efficacy for early diagnosis, risk stratification of multiple organ failure, and outcome prediction in severe injury cases involving children. Additionally, the inclusion of the S100 protein in the study allowed for further assessment of brain damage in cases of traumatic brain injury (TBI), contributing to the overall prognostic model.

## 1. Introduction

Severe injury, primarily caused by falls or road traffic accidents, is a leading cause of disability and mortality in children aged 1 and older [1]. The mortality risk in children with severe injury is closely linked to the development of multiple organ failure (MOF). The rate of post-injury MOF in children treated in an intensive care unit (ICU) ranges from 11.3% to 23.1%, with a mortality rate of 20.1% to 53% among injured children with MOF, compared to only 0.5% among those without MOF [2,3]. MOF can develop rapidly in children, with 75–86% of cases occurring within the first 24 h in the ICU [4]. The mechanisms of inflammatory reactions and immune responses in severe injury differ between children and adults, which may explain these differences [5]. In 95% of cases, MOF in children is accompanied by a severe, unregulated, systemic inflammatory response, leading to an imbalance of hyperinflammation and immunosuppression during the critical period of injury, increasing the risk of infections and septic complications (SC) such as sepsis, severe sepsis, and septic shock in the post-injury period. The development of MOF and SC cumulatively elevates the risk of mortality in children [6]. Early detection of MOF and SC significantly improves treatment outcomes. To achieve this goal, various MOF assessment scales, such as pSOFA, PELOD, PELOD-2, and MODS, have been used in pediatric practice [7]. Additionally, researchers continue to search for potential prognostically significant markers for injury severity, MOF, SC, and short-term (28-day survival) and long-term outcome prediction in severely injured children (6-month survival). Recently, markers such as Il-6 [2], procalcitonin (PCT), C-reactive protein (CRP) [8], protein S100 [9], neuron-specific enolase (NSE) [9], adrenomedullin (ADM) [10], and combinations thereof have been considered clinically available markers. Among them, ADM and its surrogate marker mid-regional proadrenomedullin (MR-proADM) have shown promising results in adult patients [11,12]. ADM, a pleiotropic regulatory peptide, is produced by various tissues in response to hypoxia, infectious agents, pro-inflammatory cytokines (IL-1b and TNF-a), angiotensin II, endothelin I, and nitric oxide (NO). Compared to ADM, MR-proADM is a more stable peptide with a longer half-life and no biological effect. MR-proADM levels reflect the concentration of ADM and allow us to determine its actual functional secretion [13,14,15]. Clinical data on the significance of ADM or MR-proADM levels in children with severe injuries for MOF prediction are lacking, as most studies have focused on adult groups. Existing published data mainly discuss the use of MR-proADM as a potential prognostic marker for the development of septic states in cases of various bacterial and viral infections [16,17]. MR-proADM did not demonstrate better diagnostic and prognostic accuracy as an independent predictor of septic conditions, compared to more common CRP and PCT biomarkers. However, for predicting 28-day mortality and MOF in adult patients, MR-proADM emerged as a potential biomarker candidate that significantly improved the quality of the predictive model, especially in serial measurements, independent of other biomarkers [12,16].

The aim of the study was to determine informative criteria of the severity and outcome prediction of injury in children based on MR-proADM, PCT, NSE, and protein S100 levels in the critical period of severe injury.

## 2. Materials and Methods

In our retrospective study, we examined 122 plasma and 122 serum samples obtained from 52 patients, consisting of 32 boys (61.5%) and 20 girls (38.5%), during the critical period of severe injury. The primary causes of injuries in children were road traffic accidents (52%, n = 27) and falls from heights (48%, n = 25). These patients were treated in the intensive care unit (ICU) of the Institute of Urgent Children Surgery and Traumatology between 2019 and 2022. Laboratory examinations were conducted in the laboratory department of the National Medical Research Center for Children’s Health, Ministry of Health of the Russian Federation, between 1 to 4 times in the post-injury period, depending on the time spent by the child in the ICU. The age of the children ranged from 6 months to 17 years, and the laboratory examination was performed on the 1st, 3rd, 7th, and 14th days of their stay in the ICU.

The patients in our study met the following criteria: severe injury (ISS ≥ 16), aged between 2 months and 18 years, and admitted to the ICU within 48 h of the injury. Patients with concomitant acute inflammatory and chronic diseases were excluded from the study. Of the total patients, 63% (n = 33) were admitted to the ICU within 24 h after the injury, and 37% (n = 19) were admitted within 48 h.

To assess injury severity, we used the Glasgow Coma Scale (GCS) and the Injury Severity Score (ISS), calculated from the Abbreviated Injury Scale (AIS), considering the three most severely injured body regions. According to the ISS definition, “severe trauma” is characterized by an ISS ≥16, which is also validated for pediatric polytrauma.

The Glasgow Coma Outcome Scale (GOS) and the Severe Injury Outcomes Scale (OISS) were used to assess the outcome of severe injury: [18]: 1—full recovery (keeping the same well-being with the same level of activity); 2—good recovery (with some consequences that do not affect the level of social adaptation, but limit the level of functional capacity, further rehabilitation needed); 3—moderate disability (with consequences that decrease functional capacity, no need for assistance in everyday life); 4—severe disability (permanent need for assistance in everyday life); 5—death. Assessments of OISS and GOS were performed at the time of the patient’s discharge.

Using OISS, patients were divided into 2 groups: severe injury with favorable outcome (OISS 1-3, n = 44) and severe injury with unfavorable outcome (OISS 4-5, n = 8). The group of severe injuries with lethal outcomes (n = 4) was described separately (Table 1). Using GOS patients with TBI were divided into 2 groups: TBI with favorable outcome (GOS 5, n = 35) and TBI with unfavorable outcome (GOS 1-4, n = 15). According to GOS, an unfavorable outcome included a lethal outcome and any level of disability.

Clinical and laboratory indicators of systemic inflammatory response syndrome and organ failure were evaluated in all patients. Organ functioning was assessed daily after admission to the ICU using MODS (Multiple Organ Dysfunction Score) [19]. According to this descriptor, a daily assessment of five organ dysfunctions (lungs, liver, kidneys, hemodynamics, and consciousness) from 0 (no dysfunction) to 4 points (severe dysfunction) is made. A value of >3 points represents organ dysfunction. MOF was diagnosed when 2 organ dysfunctions were found simultaneously. Patients were divided into groups depending on the development of septic complications (with SC (n = 16) and SC no (n = 36)) and multiple organ failure (MOF (n = 8) and MOF no (n = 44)).

The control group in the study consisted of fourteen healthy children, who underwent a medical examination at the National Medical Research Center for Children’s Health and were comparable in age and sex, with a median age of 12.0 years (interquartile range: 7.0–16.0 years) and comprising 8 boys (66.6%) and 6 girls (33.4%).

Plasma and serum samples obtained after centrifugation of patients’ blood were stored at −80 °C until testing. Plasma levels of MR-proADM, as well as serum levels of PCT, NSE, and protein S100 (S100A1B and S100 BB), were assessed. The level of MR-proADM in blood plasma and the level of PCT in blood serum were estimated by the automated immunochemical analyzer Thermo Scientific™ BRAHMS™ KRYPTOR™ (Thermo Fisher Scientific, Hennigsdorf, Germany) based on TRACE™ technology, using BRAHMS MR-proADM KRYPTOR (Brahms Kryptor^®^, Hennigsdorf, Germany, catalogue number 829050) and BRAHMS PCT sensitive KRYPTOR (Brahms Kryptor^®^, Hennigsdorf, Germany, catalogue number 825050) reagents. The manufacturer’s reference values for MR-proADM are <0.87 nmol/L (detection limit 0.05 nmol/L) and 0.064 ng/mL for PCT (detection limit 0.020 ng/mL). Serum levels of NSE and S100 were estimated on the automatic immunochemical analyzer Cobas e411 (Roche Diagnostics, Rotkreuz, Switzerland) using Roche Diagnostics Elecsys NSE (Roche Diagnostics GmbH, Mannheim, Germany, catalogue number 12133113) and Elecsys S100 (Roche Diagnostics GmbH, Mannheim, Germany, catalogue number 03175243) reagents. The manufacturer’s reference values for NSE are 16.3 ng/mL (detection limit 0.05 ng/mL) and 0.105 μg/L for S100 (detection limit 0.005 μg/L). For the control group, only the level of MR-proADM in blood plasma was evaluated.

The obtained data were processed in MS Excel 2016 (Microsoft corp., Washington, DC, USA) and Statistica10 (StatSoft, Inc., Oklahoma, OK, USA), also using the language “R” [20] with the package extension “pROC” [21]. The results are presented as median (Me) and interquartile range (Q25–Q75). The Mann–Whitney–Wilcoxon (MWW) U test was used to compare the differences. In the case of multiple comparisons, the analysis of the difference in features was carried out with an adjustment of the level of confidence considering the Bonferroni correction. Spearman’s correlation coefficient was used to measure the relationship between features. The significance of quantitative indicators was assessed by simple and multiple logistic regression. Cut-off values for the outcome prediction after injury were chosen in the groups of patients with favorable and unfavorable outcomes according to OISS and GOS, as well as depending on the development of SC and MOF on the receiver operating characteristic curve (ROC curve). The main criteria for the determination of cut-off values were maximum sensitivity and specificity. A *p* value of <0.05 (*) or < 0.01 (**) was considered statistically significant.

## 3. Results

We conducted a correlation analysis between MR-proADM, PCT, NSE, and protein S100 levels obtained from children with severe injuries on the first day in the ICU and clinical parameters (Table 2). Our analysis revealed significant correlations: blood loss degree with MR-proADM and NSE levels, SC with PCT levels, and MOF with all analyzed markers. Notably, MR-proADM exhibited the most significant correlation (*p* < 0.00001). Additionally, GOS and OISS estimates correlated with MR-proADM and protein S100 levels (Table 2).

By employing the nonparametric Mann–Whitney test, we compared MR-proADM, PCT, NSE, and protein S100 level differences in children with injuries across different time periods and groups: OISS groups (Table 3), GOS groups, with and without SC, with and without MOF. We found significant differences in MR-proADM levels in OISS groups (Table 3) from the 1st to the 7th day, in GOS groups from the 1st to the 3rd day, and in groups with and without MOF throughout the observation period. However, no significant differences were observed for SC. Significant differences in PCT levels were found in OISS, SC, and MOF groups from the 1st to the 3rd day, and in GOS groups on the 3rd day. For NSE, significant differences appeared in OISS groups on the 3rd day and in MOF groups from the 1st to the 3rd day. Moreover, a significant difference in protein S100 levels was identified in OISS, GOS, and MOF groups from the 1st to the 3rd day.

Despite considerable variability in biomarker levels among groups, patients with MOF displayed significantly higher median biomarker concentrations within 1–3 days after ICU admission compared to upper limits of normal defined by test system manufacturers, exceeding levels in patients without MOF. For the control group, the MR-proADM level was Me [IQR] 0.21 [0.16, 0.26] nmol/L, significantly lower than in severely injured children (Mann–Whitney U test, *p* = 0.006) (Figure 1).

We employed logarithmic regression to evaluate relationships between MR-proADM, PCT, NSE, and protein S100 levels in children on the first and third days after ICU admission and the outcomes of severe injury, TBI, as well as the development of SC and MOF (Table 4). Our regression analysis highlighted MR-proADM’s strongest association with MOF development and severe injury outcomes. Specifically, MOF development was more closely tied to MR-proADM values obtained on the 3rd day (OR 3740.00, 95% CI 1.34–10,400,000.00, *p* < 0.05 vs. OR 142.00, 95% CI 4.47–4480.00, *p* < 0.01 for MR-proADM_1 day_), while severe injury outcomes correlated more with MR-proADM values on the 1st day after admission to the ICU (OR 142.00, 95% CI 4.47–4480.00, *p* < 0.01 vs. OR 11.3, 95% CI 1.4 to 91.4, *p* < 0.05 for MR-proADM_3 days_). Other markers showed weaker associations with severe injury outcomes, MOF development, and SC, except for protein S100 levels, which exhibited a strong association with TBI outcomes (TBI outcome ~ S100 on the third day—OR 12.40, 95% CI 1.25–124.00, *p* <0.05) (Table 4). 

When using ROC analysis, the characteristics of the separation models based on MR-proADM, PCT, NSE, and S100 protein levels obtained on the first day after admission to the ICU were evaluated, depending on the outcome of injury in children according to OISS and GOS, as well as the development of SC and MOF (Table 5). We found that the OISS separation model for MR-proADM had excellent quality, very good quality for PCT, and good quality for S100. The GOS separation model had good quality for MR-proADM. As for SC, a separation model of good quality was obtained for PCT. For MOF, a separation model of excellent quality was obtained for MR-proADM, one of very good quality was obtained for PCT and S100, and one of good quality was obtained for NSE. The best quality of the predictive model was obtained for MOF. The optimal cut-off points for MOF prediction were for MR-proADM at 0.929 nmol/L (Se 93.2%, Sp 87.5%; AUC: 0.963 95%CI 0.911–1), for PCT at 4.20 ng/mL (Se 77.3%, Sp 75.0%; AUC: 0.864 95%CI: 0.739–0.988), for S100 at 0.493 μg/L (Se 84.1%, 87.5%; AUC: 0.830 95%CI: 0.6–1), and for NSE at 54.45 ng/mL (Se 75.0%, Sp 87.5%; AUC: 0.791 95%CI: 0.578–1) (Table 5, Figure 2).

Using multiple logistic regression analysis, including and excluding biomarkers stepwise, we found significant ones for predicting the outcome of OISS, TBI, and the development of SC and MOF. We also estimated the diagnostic characteristics of the obtained models using ROC analysis:OISS outcome ~ MR-proADM_1 day_ and PCT_3 day_ (AUC: 0.994 95% CI 0.978–1),GOS outcome ~ S100_3 days_ (AUC: 0.836 95% CI 0.646–1),SC ~ MR-proADM_1 day_, PCT_3 day_, NSE_3 day_, S100_3 day_ (AUC: 0.9 95% CI 0.795–1),MOF ~ MR-proADM_3 day_ and PCT_1 day_ (AUC: 0.999 95% CI 0.999–1).

A comprehensive serial assessment of biomarkers has significantly improved the quality of separation models for the prediction of the outcome of OISS, GOS, and the development of SC and MOF.

## 4. Discussion

The aim of our retrospective study was to investigate whether the level of MR-proADM, which directly reflects the concentration of ADM, can provide additional clinically relevant information on the pathophysiology of injury, post-injury MOF, and the mechanisms of SC development in cases of severe injury in children. We also examined the correlation between MR-proADM and well-established biomarkers such as PCT, NSE, and protein S100. While ADM was originally thought to have only vasodilating properties [22], recent research has shown that it also plays a role in modulating inflammation and regulating vascular tone and endothelial permeability [23]. In the acute systemic inflammatory response, ADM helps reduce vascular permeability and maintains the stability and integrity of the endothelium [24]. Endotheliopathy caused by traumatic shock can severely affect the outcome of injury during the critical period [25], and endothelial dysfunction is considered a major cause of multiple organ failure in cases of infection, severe injury, burns, or serious surgical interventions [24,26]. Patients in critical condition had very high levels of MR-proADM, and extremely high levels were found in patients with septic conditions and multiple organ failure. MR-proADM also showed a strong correlation with systemic inflammation markers such as CRP, IL6, and TNFα, as well as significant associations with endothelial dysfunction biomarkers (SDMA, ADMA, and C-CTproET1) [27].

Our study found that within 72 h after admission to the ICU, the median concentrations of the analyzed biomarkers were significantly higher in patients with MOF compared to patients without MOF. This was especially true for MR-proADM and PCT. MR-proADM level showed the strongest association with MOF and the outcome of severe injury. Other markers showed weaker associations with the outcome of severe injury, MOF, and SC. Protein S100 was strongly associated with GOS and TBI outcome, while NSE level correlated with the degree of blood loss. Scientific data suggest that increased NSE levels are associated with hypoxia and cerebral ischemia in traumatic injury, and in combination with S100, they indicate the degree of impairment of the blood–brain barrier [28].

For the control group, the level of MR-proADM was ME [IQR] 0.21 [0.16, 0.26] nmol/L, which was significantly lower than in children with severe injury. According to scientific data, reference values of MR-proADM in the plasma of adult healthy donors were Mean [95% CI] 0.33 [0.17–0.49] nmol/L [15]; comparable values were obtained for serum level of MR-proADM, i.e., Mean [95%CI] 0.36 [0.27–0.51] nmol/L [14], with a direct association of MR-proADM concentrations with age [13]. For this reason, our MR-proADM values for the control group were slightly lower in adult volunteers.

We also aimed to determine the optimal cut-off point of MR-proADM to identify patients at high risk of septic conditions and MOF. During the study, we found the following optimal cut-off points for MOF prediction after admission to the ICU: MR-proADM—0.929 nmol/L (Se 93.2%, Sp 87.5%), PCT—4.20 ng/mL (Se 77.3%, Sp 75.0%), S100—0.493 μg/L (Se 84.1%, Sp 87.5%), NSE—54.45 ng/mL (Se 75.0%, Sp 87.5%). The cut-off point of MR-proADM stated by the test system manufacturer was 0.87 nmol/L. Optimal cut-off points of MR-proADM for diagnosing sepsis in adult patients were reported to be 1.0–1.5 nmol/L [29,30]. An MR-proADM level of 1.5–2.0 nmol/L was found to be an independent predictor of MOF-related mortality in adult critically ill patients [27,31]. The cut-off level of MR-proADM for diagnosing kidney damage in urinary tract infections [32] in children was 0.66 nmol/L, and 0.70 nmol/L was closely related to ICU hospitalization [33]. Our MR-proADM level of 0.929 nmol/L was slightly higher than reported by the manufacturer and other studies of children [32,33], which may be attributed to the severity of the patients’ illnesses.

None of the biomarkers can adequately reflect the conditions of a patient at risk of developing SC and MOF, so some researchers have proposed using a combination of biomarkers to increase diagnostic and prognostic accuracy. In a retrospective study involving 104 patients with sepsis, the combination of PCT, MR-proADM, and TNFα showed the best results in early sepsis detection compared to individual biomarkers [34]. However, in another study, the diagnostic accuracy of MR-proADM for sepsis detection was inferior to clinically used biomarkers (CRP, PCT, and IL-6), and the combination of MR-proADM with any of these markers did not improve diagnostic accuracy. Nevertheless, MR-proADM levels were strongly associated with organ dysfunction and overall mortality [16,27]. The predictive potential of biomarkers is significantly higher with serial measurements. For example, MR-proADM showed a 42% increase in prognostic significance for sepsis detection when measured again 72 h after admission to the ICU [11]. In our study, the combination of MR-proADM and PCT showed the greatest diagnostic and prognostic accuracy for MOF and the outcome prediction of severe injury. We also identified the most informative time intervals for the evaluation of these markers. For MOF prediction, a consistently high level of MR-proADM of more than 0.929 nmol/L for 72 h and a PCT level of more than 4.20 ng/mL upon admission to the ICU were important. Conversely, for the outcome prediction of severe injury, the initial level of MR-proADM of 0.929 nmol/L and the dynamics of PCT in the post-injury period were important. A comprehensive serial assessment of biomarker levels—MR-proADM on admission and PCT, NSE, and S100 for 72 h—significantly improved the diagnostic and prognostic effectiveness of the model for predicting the development of septic complications.

In cases of TBI, the continuing high level of protein S100 was the most informative for outcome prediction using GOS. Since TBI can result in long-term cognitive changes or deficits, it is also important to explore brain biomarkers that may help to predict these differences. Many aspects of the central nervous system (CNS) in the pediatric population, e.g., myelination and synapse formation, are in continual development, and brain injury in children could severely impact these brain maturation processes with lasting neurological consequences [35]. Chang et al. found that risks of attention-deficit/hyperactivity disorder (ADHD), autism spectrum disorder (ASD), and developmental delay (DD) increased after severe TBI compared with mild and moderate TBI, especially for children who have had TBI under the age of three years [36]. Mounting evidence indicates age-dependent differences in the pathophysiology of neurological diseases necessitating increased use of immature animals when modeling pediatric TBI [37]. 

## 5. Conclusions

MR-proADM has most of the characteristics of the “ideal” marker in the SMART concept: “S”—specific and sensitive; “M”—measurable; “A”—available and affordable; “R”—responsive and reproductive; “T”- timely for screening, diagnosing, stratifying the risk of MOF, and, ultimately, predicting the outcome of severe injury. The availability of standardized automated test systems for the estimation of MR-proADM can ensure its widespread adoption in clinical practice. MR-proADM can be evaluated both independently and in combination with other markers of inflammation, damage, and dysfunction of the immune system, as well as in conjunction with the internal systems (scales) of assessing the severity of patients. The use of such a multimodal approach to early diagnosis and prediction allows for identifying life-threatening conditions such as sepsis and MOF more accurately. And it significantly improves survival after severe injury.

Our study has some obvious limits, including the monocentric design, a relatively small patient sample, and an unbalanced sample. It would be useful to conduct additional studies to further validate the role of both MR-proADM and PCT in severe trauma pediatric patients’ care in the emergency department.

## Figures and Tables

**Figure 1 biomedicines-11-02306-f001:**
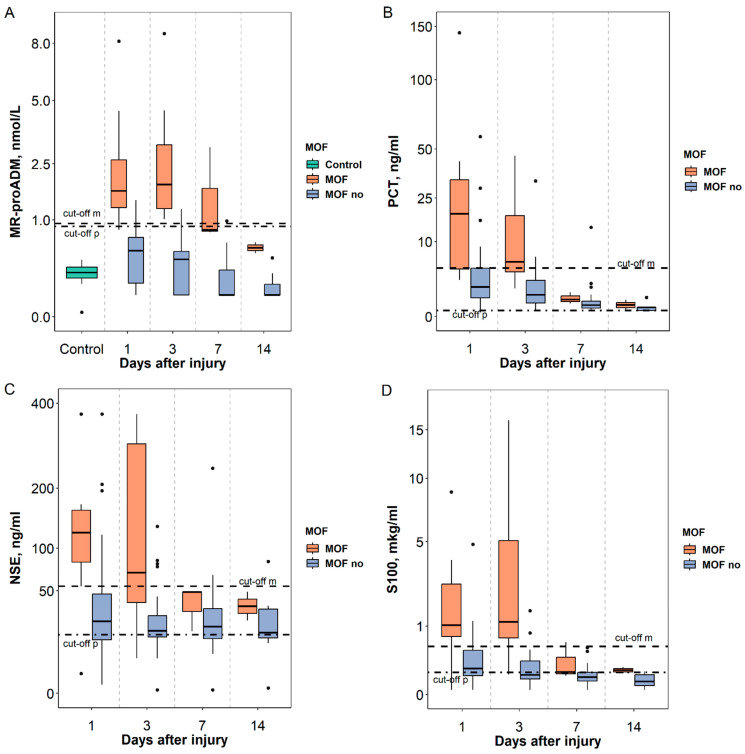
MR-proADM (**A**), PCT (**B**), NSE (**C**), and protein S100 (**D**) levels in children with injury over time, depending on MOF development. Note: Me [Q25%–Q75%] (Min-Max), dots indicate emissions; cut-off model threshold marked in dotted line, cut-off product manufacturer threshold marked in dash-dotted line. Comparison groups: with and without MOF, for MR-proADM: Control Group (CG), with and without MOF.

**Figure 2 biomedicines-11-02306-f002:**
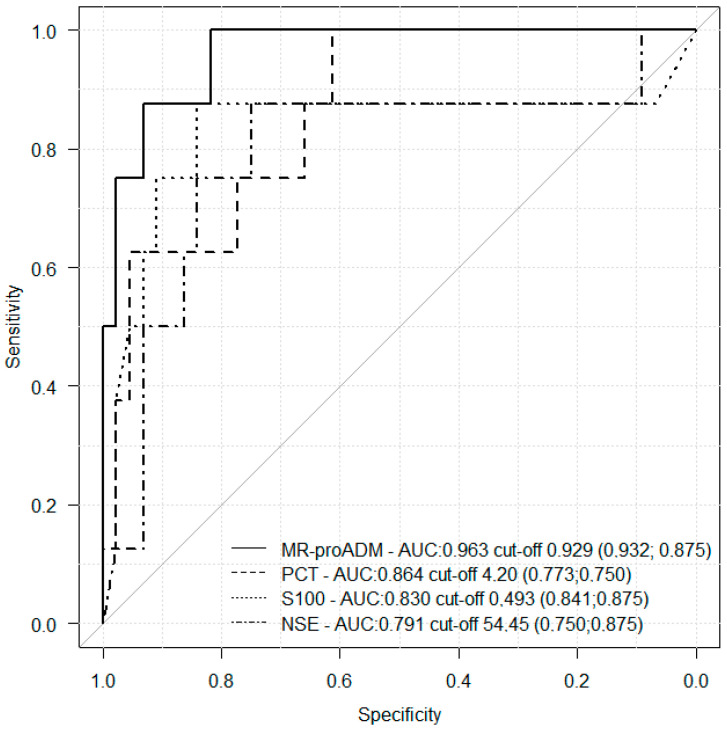
Characteristic ROC curves of MR-proADM, PCT, NSE, and protein S100 on the first day after admission to the ICU, depending on the development of MOF in children with severe injury.

**Table 1 biomedicines-11-02306-t001:** Clinical characteristics of patients.

Factor		OISS		
		1-3	4-5	5
n		44	8	4
Gender (%)	f	19 (43.2)	1 (12.5)	1 (25.0)
	m	25 (56.8)	7 (87.5)	3 (75.0)
Age (Me [IQR]), years		12.50 [6.75, 15.00]	12.00 [3.50, 13.00]	13.00 [10.25, 13.50]
ICU days (Me [IQR]), days		12.00 [7.00, 16.25]	17.50 [7.50, 34.50]	7.00 [6.00, 9.25]
Total bed days (Me [IQR]), days		28.00 [19.00, 48.25]	33.00 [7.50, 57.25]	7.00 [6.00, 9.25]
ISS (Me [IQR])		26.00 [21.00, 29.00]	28.00 [25.00, 34.25]	32.00 [28.00, 35.75]
GCS (Me [IQR])		10.50 [6.75, 13.00]	4.50 [3.75, 8.25]	4.00 [3.75, 5.25]
Coma (%)		18 (40.9)	8 (100.0)	4 (100.0)
Combined trauma (%)		39 (88.6)	6 (75.0)	3 (75.0)
Multiple trauma (%)		22 (50.0)	3 (37.5)	2 (50.0)
Open trauma (%)		19 (43.2)	6 (75.0)	4 (100.0)
TBI (%)		42 (95.5)	8 (100.0)	4 (100.0)
AIS TBI (%)	0	2 (4.5)	-	-
	1	8 (18.2)	-	-
	2	6 (13.6)	-	-
	3	11 (25.0)	3 (37.5)	-
	4	13 (29.5)	-	-
	5	4 (9.1)	5 (62.5)	4 (100.0)
GOS (%)	No TBI	2 (4.5)	-	-
	TBI favorable outcome	35 (79.5)	-	-
	TBI unfavorable outcome	7 (15.9)	8 (100.0)	4 (100.0)
Blood loss (%)		29 (65.9)	7 (87.5)	3 (75.0)
Blood loss degree (%)	0	15 (34.1)	1 (12.5)	1 (25.0)
	1	9 (20.5)	1 (12.5)	-
	2	14 (31.8)	2 (25.0)	2 (50.0)
	3	6 (13.6)	4 (50.0)	1 (25.0)
Vasopressor support (%)		24 (54.6)	7 (87.5)	4 (100.0)
Unstable hemodynamics (%)		21 (47.7)	7 (87.5)	4 (100.0)
Ventilator (%)		37 (84.1)	8 (100.0)	4 (100.0)
Complications (%)		12 (27.3)	8 (100.0)	4 (100.0)
SC (%)		11 (25.0)	5 (62.5)	2 (50.0)
MOF (%)		2 (4.5)	6 (75.0)	4 (100.0)

**Table 2 biomedicines-11-02306-t002:** Spearman’s R correlation coefficient for MR-proADM, PCT, NSE, and protein S100 levels and clinical indicators on the 1st day of hospitalization in the ICU.

Index	Factor	N	Spearman R	t(N-2)	*p*-Level
MR-proADM	Blood loss degree	52	0.40	3.12	<0.01
	MOF	52	0.58	5.05	<0.00001
	OISS	52	0.46	3.62	<0.001
	GOS	52	0.39	3.02	<0.01
PCT	SC	52	0.46	3.67	<0.001
	MOF	52	0.45	3.61	<0.001
S100	Coma	52	0.31	2.33	<0.05
	GCS	52	−0.28	−2.10	<0.05
	MOF	52	0.41	3.20	<0.01
	OISS	52	0.36	2.71	<0.01
	GOS	52	0.31	2.34	<0.05
NSE	Blood loss degree	52	0.35	2.66	<0.05
	MOF	52	0.36	2.76	<0.01

**Table 3 biomedicines-11-02306-t003:** Dynamics of MR-proADM, PCT, NSE, and protein S100 levels in the critical period of severe injury in groups with favorable and unfavorable outcomes according to OISS.

Parameter		OISS Favorable Outcome	OISS Unfavorable Outcome	*P_adj_*-Level
MR-proADM, nmol/L	1d	0.47 [0.05, 2.11] ***	1.43 [0.82, 8.11] ***	<0.001
	3d	0.36 [0.05, 2.02] **	1.60 [0.61, 8.57] **	0.004
	7d	0.05 [0.05, 0.81] *	0.98 [0.34, 3.07] *	0.048
	14d	0.05 [0.05, 0.43]	0.48 [0.37, 0.59]	0.216
NSE, ng/ml	1d	26.89 [0.36, 370.00]	81.08 [1.81, 370.00]	0.452
	3d	18.85 [0.05, 132.10]	150.74 [5.85, 370.00]	0.152
	7d	21.09 [0.05, 240.50]	31.30 [12.82, 48.70]	3.364
	14d	17.49 [0.12, 82.45]	28.14 [25.12, 31.16]	3.080
PCT, ng/ml	1d	1.55 [0.03, 42.99] *	18.80 [2.36, 143.40] *	0.012
	3d	1.01 [0.06, 6.37] **	18.22 [1.41, 46.05] **	0.008
	7d	0.24 [0.05, 1.93]	1.05 [0.10, 14.13]	1.112
	14d	0.07 [0.05, 0.16]	0.57 [0.50, 0.64]	0.160
S100, μg/L	1d	0.15 [0.00, 4.82] *	0.85 [0.00, 8.77] *	0.048
	3d	0.08 [0.00, 1.50] **	1.50 [0.20, 16.07] **	0.004
	7d	0.08 [0.00, 0.47]	0.08 [0.04, 0.58]	2.408
	14d	0.04 [0.00, 0.16]	0.10 [0.10, 0.11]	1.520

Note: Me [Q25%–Q75%]; Mann–Whitney U test; *P_adj_*—adjusted level of reliability of the analyzed parameters considering the Bonferroni correction; * *p* < 0.05; ** *p* < 0.01; *** *p* < 0.001; comparison groups: favorable outcome OISS, unfavorable outcome OISS.

**Table 4 biomedicines-11-02306-t004:** Relationship of MR-proADM, PCT, NSE, and protein S100 levels in the 1st and 3rd days after admission to the ICU with outcome prediction of severe injury, TBI, and the development of SC and MOF.

1 d ICU	MR-proADM	PCT	NSE	S100
OISS				
OR	34.40	1.09	1.01	1.61
95%CI	2.97–398.00	1.00–1.10	0.99–1.02	0.96–2.69
*p*-level	0.004 **	0.015 *	0.067	0.066
χ2Pr, *p*-level	0.000002 ***	0.00076 ***	0.067	0.033 *
GOS				
OR	1.46	6.57	0.99	93.10
95%CI	0.12–16.5	0.21–205.0	0.97–1	0.001–8,130,000.00
*p*-level	0.760	0.284	0.059	0.435
χ2Pr, *p*-level	0.695	0.04134 *	0.075	0.212
SC				
OR	0.92	0.98	1.00	0.86
95%CI	0.59–1.45	0.96–1.01	0.99–1.01	0.58–1.28
*p*-level	0.744	0.383	0.632	0.472
χ2Pr, *p*-level	0.747	0.366	0.635	0.472
MOF				
OR	142.00	1.07	1.01	2.30
95%CI	4.47–4480.00	1.01–1.14	1.00–1.02	1.07–4.92
*p*-level	0.004 **	0.025 *	0.021 *	0.032 *
χ2Pr, *p*-level	0.0000001 ***	0.002 **	0.012 *	0.002 **
**3 d ICU**	**MR-proADM**	**PCT**	**NSE**	**S100**
OISS				
OR	11.3	1.40	1.02	20.9
95%CI	1.4–91.4	0.98–2.20	1.00–1.04	1.76–248.0
*p*-level	0.0227 *	0.061	0.035 *	0.016 *
χ2Pr, *p*-level	0.0001 ***	0.0003 ***	0.0001 ***	0.00006326 ***
GOS				
OR	4.70	1.43	1.03	12.40
95%CI	0.97–23.10	0.96–2.11	0.99–1.06	1.25–124.00
*p*-level	0.054	0.070	0.057	0.031 *
χ2Pr, *p*-level	0.002 **	0.0003 ***	0.0008702 ***	0.0005 ***
SC				
OR	0.90	1.03	1.01	1.04
95%CI	0.55–1.57	0.96–1.11	0.99–1.01	0.83–1.30
*p*-level	0.795	0.389	0.199	0.722
χ2Pr, *p*-level	0.787	0.389	0.183	0.726
MOF				
OR	3740.00	1.11	1.02	15.8
95%CI	1.34–10,400,000.00	0.99–1.23	1.00–1.04	1.60–157.00
*p*-levels.	0.0422 *	0.053	0.048 *	0.018 *
χ2Pr, *p*-level	0.0000001 ***	0.015 *	0.001311 **	0.0001315 ***

Note: OR—odds ratio; 95%CI—95% confidence interval; χ2P is Pearson’s chi-squared test; * *p* < 0.05; ** *p* < 0.01; *** *p* < 0.001.

**Table 5 biomedicines-11-02306-t005:** ROC analysis for MR-proADM, PCT, NSE, and protein S100 obtained on the first day after admission to the ICU to predict the outcome of severe injury in children by OISS and GOS and the development of SC and MOF.

	MR-proADM	PCT	NSE	S100
OISS				
cut-off	0.929	18.55	54.45	0.413
Se	93.2	95.5	72.7	72.7
Sp	87.5	62.5	75.0	75.0
AUC	0.960 ***	0.838 **	0.678	0.778 *
95%CI	0.908–1	0.690–0.986	0.431–0.924	0.552–1
GOS				
cut-off	0.819	2.23	54.45	0.312
Se	85.7	54.3	77.1	68.6
Sp	66.7	73.3	60.0	73.3
AUC	0.791 **	0.663	0.632	0.677
95%CI	0.640–0.943	0.486–0.840	0.445–0.820	0.498–0.856
SC				
cut-off	0.609	2.364	32.04	0.196
Se	69.4	66.7	66.7	61.1
Sp	68.8	87.5	62.5	75.0
AUC	0.665	0.788 *	0.570	0.627
95%CI	0.489–0.841	0.661–0.916	0.394–0.747	0.465–0.789
MOF				
cut-off	0.929	4.20	54.45	0.493
Se	93.2	77.3	75.0	84.1
Sp	87.5	75.0	87.5	87.5
AUC	0.963 ***	0.864 **	0.791 *	0.830
95%CI	0.911–1	0.739–0.988	0.578–1	0.6–1 **

Note: cut-off—threshold value of the separation model; Se—sensitivity; Sp—specificity; 95%CI—95% confidence interval; AUC—area under the curve; *** AUC (0.9–1)—excellent model quality; ** AUC (0.9–0.8)—very good quality of the model; * AUC (0.8–0.7)—good quality model.

## Data Availability

The data presented in this study are available upon reasonable request from the corresponding author.

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
