# Peer review of "Comprehensive Assessment of Mid-Regional Proadrenomedullin, Procalcitonin, Neuron-Specific Enolase and Protein S100 for Predicting Pediatric Severe Trauma Outcomes"

_biomedicines, 2023, doi:10.3390/biomedicines11082306_

Round 1

Reviewer 1 Report

The authors of the article Diagnostic and prognostic value of midregional proadrenomedullin in the critical period of severe injury in children have measured proadrenomedullin (MR-14 proADM), procalcitonin (PCT), NSE and S100 protein levels in 52 children with severe injury (ISS≥16), at 4 time points up to 14 days after injury. The study included 14 healthy kids. The injured patients were further divided into groups with favorable and unfavorable outcomes with different number of patients per group if OISS or GOS scales were used. They have concluded that MR-proADM and PCT provide the highest diagnostic and prognostic value for early diagnosis and outcome of severe injury in children. The inclusion of S100 protein allowed for further assessment of brain damage in case of TBI. The authors conclude that MR-proADM measured both independently and in combination with other markers of inflammation could be used to assess the severity of critically ill patients under the age of 18.

Comments:

1.      The study examined small group of the patients admitted in children ICU. The cause of injuries was different and only TBI injury was separated from other injury causes.

2.      The study included 44 patients with favorable outcome and only 8 patients with unfavorable outcome, which make conclusions and predictions biased. It also seems, based on Table 1, that all patients with unfavorable outcome had TBI. Other exact causes were not mentioned while development of sepsis contributed considerably to unfavorable outcome. Patient were also divided by presence of MOF, which was another important factor causing unfavorable outcome. Overall, the observation about importance of measuring MR-proADM and PCT in seriously injured patient is relevant but it is not novel. S100B change is also known as important predictor of TBI severity.

3.      The study had control group of healthy kids, which is advisable group to have for reference values. However, in this case, it would be much more important to have a group of children with less severe injuries (ISS < 16), admitted to the hospital, but not in ICU. As this study is retrospective studies, it is possible to add this group and make conclusions much more relevant. In addition, adding more cases in ICU group would be strongly recommended and make statistics more powerful and conclusions more relevant.  

4.      No study limitations have been listed.

5.      Parts of discussion, about the pathophysiology role of measured factors, especially MR-proADM, belongs to the Introduction. Also, reference and cut-off values belong to the Method section, not discussion. Discussion with results from other studies should be expended.

6.      English should be improved considerably, and Russian words removed from the manuscript (as in raw 72).

Moderate English editing is necessary. 

Author Response

Point 1:

The study examined small group of the patients admitted in children ICU. The cause of injuries was different and only TBI injury was separated from other injury causes.

Response 1:

We are grateful and appreciate your time for sharing your rating with us.

The Institute of Urgent Children Surgery and Traumatology is a level-I pediatric trauma center, specializing in treating the most severe cases of trauma. Since severe trauma in children is rare, even large trauma centers may only encounter a small number of such cases in a year. It took us 3 years to gather an adequate sample of patients meeting our inclusion and exclusion criteria. In our study, the primary causes of injuries in children were road traffic accidents (52%, n=27) and falls from heights (48%, n=25). The most commonly injured organs in children were the head and thorax, followed by extremities and abdominal injuries. Traumatic Brain Injuries (TBIs) were observed in 95.5% of trauma cases (n=50), with TBI being the dominant injury type (AIS>=4) in 42% of cases (n=22). We analyzed patients with TBI (n=50) in groups based on their score on the Glasgow Outcome Scale. To assess injury severity, we used the Injury Severity Score (ISS), calculated from the Abbreviated Injury Scale (AIS), considering the three most severely injured body regions. According to the ISS definition, "severe trauma" is characterized by an ISS ≥16, which is also validated for pediatric polytrauma. A detailed summary of patients' clinical characteristics is provided in Table 1.

Point 2:

The study included 44 patients with favorable outcome and only 8 patients with unfavorable outcome, which make conclusions and predictions biased. It also seems, based on Table 1, that all patients with unfavorable outcome had TBI. Other exact causes were not mentioned while development of sepsis contributed considerably to unfavorable outcome. Patient were also divided by presence of MOF, which was another important factor causing unfavorable outcome. Overall, the observation about importance of measuring MR-proADM and PCT in seriously injured patient is relevant but it is not novel. S100B change is also known as important predictor of TBI severity.

Response 2:

By collectively evaluating these indicators, we aim to develop a more accurate model for predicting the development of Multi-Organ Failure (MOF) and Septic Complications (SC) in children with severe trauma. The occurrence of MOF and SC increases the cumulative risk of unfavorable injury outcomes. Patients were classified into groups depending on the development of SC and MOF: with SC (n=16) and without SC (n=36); with MOF (n=8) and without MOF (n=44). Also we divided patients into two groups based on OISS (n=52): severe injury with a favorable outcome (n=44) and severe injury with an unfavorable outcome (n=8). Similarly, for patients with TBI, we created two groups based on GOS (n=50): TBI with a favorable outcome (n=35) and TBI with an unfavorable outcome (n=15). The comprehensive serial assessment of biomarkers significantly enhanced the predictive models for outcome assessment, using OISS (Injury Outcome Severity Scale) and GOS (Glasgow Outcome Scale).

Point 3:

The study had control group of healthy kids, which is advisable group to have for reference values. However, in this case, it would be much more important to have a group of children with less severe injuries (ISS < 16), admitted to the hospital, but not in ICU. As this study is retrospective studies, it is possible to add this group and make conclusions much more relevant. In addition, adding more cases in ICU group would be strongly recommended and make statistics more powerful and conclusions more relevant. 

Response 3:

Our experience with these markers shows that the definition of these markers is less critical for patients with mild to moderate trauma compared to patients with severe trauma treated in the ICU. Patients with severe trauma have a significantly higher risk of complications such as sepsis or multiple organ failure. According to other authors, the rate of post-injury multiple organ failure (MOF) in children treated in the ICU ranges from 11.3% to 23.1%. Moreover, the mortality rate among injured children with MOF is reported to be between 20.1% and 53%, in contrast to only 0.5% among patients without MOF. In our study, MOF developed in 15% of patients, and the mortality rate among injured children with MOF was 50%.

Point 4:

No study limitations have been listed.

Response 4:

This is a crucial point, and to enhance the manuscript, we have included research limitations.

Point 5:

Parts of discussion, about the pathophysiology role of measured factors, especially MR-proADM, belongs to the Introduction. Also, reference and cut-off values belong to the Method section, not discussion. Discussion with results from other studies should be expended.

Response 5:

We believe it is appropriate to discuss reference intervals in the discussion section, as we obtained different results compared to other studies. However, it is important to note that there are limited studies reporting on the test accuracy of MR-proADM in pediatric practice, in mainly, for the diagnosis of bacterial or viral (COVID-19) infections and/or sepsis. To provide a more comprehensive understanding of the role of MR-proADM in diagnosing multiple organ failure as a marker of endothelial dysfunction, we felt it necessary to include a brief introduction about the pathophysiological role of adrenomedullin at the beginning of the discussion section. In cases of severe trauma, the development of endothelial dysfunction is a key factor contributing to multiple organ failure. The discussion should primarily focus on the parameters of the constructed models for classifying patients with severe trauma. Given the limited number of studies on MR-proADM in pediatric practice, we believe our discussion is sufficient.

Reviewer 2 Report

The use of biomarkers such as MRproADM to predict the outcome in intensive care unite patients is of interest.

There are, however, major criticisms that should be addressed in order to render this article suitable for pubblication.

First, the study population is not clearly defined and characterized. What does severe injury means? If the majority of patients had traumatic brain injury, then the aim would be to predict brain damage? 

Second, there are two Tables indicated as 3 and reported data appear somewhat redundant and only significant results should be more clearly presented. The same for Table 5 concerning ROC curve analysis (e.g less decimals should be included, CI should be reported once in the legend, etc etc).

Third, the section related to comprehensive serial assessments of biomarkers is difficult to understand and should be entirely revised.

Last, in the discussion the outcome to be predicted by biomarkers should be definitively identified. 

English should be revised.

Author Response

Response to Reviewer 2 Comments

Point 1:

First, the study population is not clearly defined and characterized. What does severe injury means? If the majority of patients had traumatic brain injury, then the aim would be to predict brain damage?

Response 1:

We are grateful and appreciate your time for sharing your rating with us.

In the assessment of injury severity, we used the Injury Severity Score (ISS), which is calculated based on the Abbreviated Injury Scale (AIS) and considers the three most severely injured body regions. According to the ISS definition, "severe trauma" is defined as an ISS ≥16. This definition is also validated for pediatric polytrauma. The total clinical characteristics of patients are presented in Table 1.

The Institute of Urgent Children Surgery and Traumatology is a level-I pediatric trauma center, and as a result, the most severe cases of trauma are concentrated in our center. The majority of patients in need of treatment in the ICU have severe trauma or polytrauma. In our study, the primary causes of injuries in children were road traffic accidents (52%, n=27) and falls from heights (48%, n=25). The most frequently injured organs in children are the head and thorax, followed by the extremities and abdominal injuries. Our study identified varying severity of Traumatic Brain Injuries (TBIs) in 95.5% of trauma cases (n=50), with TBI being the dominant injury in the injury structure in 42% of cases (n=22). An analysis was conducted for patients with TBI (n=50) in groups, depending on their score on the Glasgow Outcome Scale.

Point 2:

Second, there are two Tables indicated as 3 and reported data appear somewhat redundant and only significant results should be more clearly presented. The same for Table 5 concerning ROC curve analysis (e.g less decimals should be included, CI should be reported once in the legend, etc etc).

Response 2:

We have made improvements to tables 2-5 to make them clearer. However, we have not removed any statistical indicators, as doing so may result in incomplete estimates of the obtained models.

Point 3:

Third, the section related to comprehensive serial assessments of biomarkers is difficult to understand and should be entirely revised.

Response 3:

For this reason, data related to models built using multiple logistic regression were presented as simply as possible. We limited ourselves to indicating the incoming components and parameters reflecting the quality of the constructed models.

Point 4:

Last, in the discussion the outcome to be predicted by biomarkers should be definitively identified.

Response 4:

By jointly evaluating these indicators, we can, first of all, build a more accurate model for predicting the development of MOF (Multi-Organ Failure) and SC (Septic Complications) in severe trauma in children. The development of MOF and SC leads to a cumulative increase in the risk of unfavorable outcomes of injury. A comprehensive serial assessment of biomarkers has significantly improved the quality of separation models for predicting outcomes using OISS (Injury Outcome Severity Scale) and GOS (Glasgow Outcome Scale).

Reviewer 3 Report

Authors tried to investigate the diagnostic and prognostic performance of MR-proADM in severe injury of children.

The following issue is critical.

They studied and compared MR-proADM, PCT, NSE, and S100. I could not understand why authors chose these various biomarkers. PCT is a well-known biomarker for bacterial infection. NSE and S100 is well-known brain injury biomarker. What they expected from the study of these biomarkers?

If they would like to investigate the role of MR-proADM in injury of children, I recommend they focus on MR-proADM, rather than mixed-biomarker study without scientific background. 

Author Response

We are grateful and appreciate you for taking the time to share your rating with us.

The aim of our work was to assess the level of MR-proADM both independently and in combination with other markers of inflammation and damage, such as PCT, NSE, and protein S100. These markers were carefully chosen based on their relevance and significance.

Procalcitonin (PCT) is a potential valid indicator for pediatric trauma, as it has shown a strong correlation with injury severity in adults. Even in the absence of infection, trauma can elevate PCT levels. Furthermore, PCT has been found to be correlated with the development of post-traumatic sepsis and can serve as a strong predictor for the development of MOF (multiple organ failure) after trauma. In children, studies have also described PCT as an independent predictor for the development of sepsis and the systemic inflammatory response syndrome (SIRS) after trauma. Plasma PCT levels have been observed to correlate with the severity of injury in children.

Additionally, other markers of organ injury, such as protein S100 and NSE, provide valuable clinical tools to identify and estimate the severity of traumatic brain injury (TBI), complementing neuroimaging techniques. The most frequently injured organs in children are the head and thorax, followed by the extremities and abdominal injuries. In our study, we detected TBIs of varying severity in 95.5% of trauma cases (n=50), and TBI was the leading cause of damage in 42% of trauma cases (n=22).

By jointly assessing these indicators, we can build a more accurate model for predicting the outcome of trauma in children. A comprehensive serial assessment of biomarkers has significantly improved the quality of separation models for the prediction of outcomes such as OISS (Outcome Injury Severity Scale), GOS (Glasgow Outcome Scale), and the development of SC (systemic complications) and MOF. Furthermore, the advantage of using this combination of biomarkers is the availability of standardized automated test systems, which is especially crucial for patients undergoing treatment in the ICU.

Your feedback prompted us to revise the title of the work to: "Comprehensive Assessment of Biomarkers for Predicting Pediatric Severe Trauma Outcomes."

Round 2

Reviewer 1 Report

The authors have answered all my questions and concerns and improved the paper considerably. I have no further comments or questions for the authors.

Minor editing of English is necessary!

Author Response

We are grateful and appreciate your time for sharing your rating with us.

We have tried to improve English language of our manuscript.

Reviewer 2 Report

Thank you for your efforts to address my criticisms.

Only minor revision needed.

Author Response

(The authors gave the same response as above.)
